# Prediction of cross-border spread of the COVID-19 pandemic: A predictive model for imported cases outside China

Ying Wang[1,2,3☯], Fang Yuan[1☯], Yueqian Song[1☯], Huaxiang Rao [ID][3], Lili Xiao[1], Huilin Guo[1], Xiaolong Zhang[1], Mufan Li[2], Jiayu Wang[2], Yi zhou Ren[2], Jie Tian[1]*, Jianzhou Yang [ID][3]*

**1** Science and Technology Research Center of China Customs, Beijing, China, **2** School of Epidemiology and Public Health, Shanxi Medical University, Taiyuan, China, **3** Department of Preventive Medicine, Changzhi Medical College, Changzhi, China

☯ These authors contributed equally to this work.
* jzyang@aliyun.com (JY); tianjie790808@163.com (JT)

## Abstract

The COVID-19 pandemic has been present globally for more than three years, and cross-border transmission has played an important role in its spread. Currently, most predictions of COVID-19 spread are limited to a country (or a region), and models for cross-border transmission risk assessment remain lacking. Information on imported COVID-19 cases reported from March 2020 to June 2022 was collected from the National Health Commission of China, and COVID-19 epidemic data of the countries of origin of the imported cases were collected on data websites such as WHO and Our World in Data. It is proposed to establish a prediction model suitable for the prevention and control of overseas importation of COVID-19. Firstly, the SIR model was used to fit the epidemic infection status of the countries where the cases were exported, and most of the r2 values of the fitted curves obtained were above 0.75, which indicated that the SIR model could well fit different countries and the infection status of the region. After fitting the epidemic infection status data of overseas exporting countries, on this basis, a SIR-multiple linear regression overseas import risk prediction combination model was established, which can predict the risk of overseas case importation, and the established overseas import risk model overall P <0.05, the adjusted R2 = 0.7, indicating that the SIR-multivariate linear regression overseas import risk prediction combination model can obtain better prediction results. Our model effectively estimates the risk of imported cases of COVID-19 from abroad.

## Introduction

Coronavirus disease 2019 (COVID-19), formerly known as novel coronavirus pneumonia, is an infectious pneumonia caused by emerging pathogens novel coronavirus ("SARS-CoV-2"); the primary transmission channels include droplets, contact, and aerosols under specific conditions [1]. Since the outbreak of COVID-19 in December 2019, 762,200,000 cases of infection

**Funding:** This study was funded by the Research Program of the General Administration of Customs of China (Grant Number, 2021HK147) and the Research and Development Project of Non-contact Health Quarantine Supervision Technology Equipment and Systems of Entry Vehicles (Grant Number 2022HK144), as well as Four "Batches" Innovation Project of Invigorating Medical through Science and Technology of Shanxi Province(Grant Number, 2022XM45). The funders had no role in study design, data collection and analysis, decision to publish, or preparation of the manuscript.

**Competing interests:** The authors declare that they have no competing interests.

and 6,890,000 cases of deaths have been caused as of April 6, 2023 [2], which seriously affects the lives and health, lifestyles, economic activities, and social order of people across the world.

Zhan C predicted the development of the epidemic and concluded that high COVID-19 contagion rates [3] and new SARS-CoV-2 variants are one of the main factors leading to multiple waves of the pandemic [4]. In the face of a highly contagious and highly mutable virus, if a country's public health system lacks preparation to deal with its cross-border spread, or if quarantine management and isolation measures for imported cases are underdeveloped, the risk of COVID-19 spread will Increase [5]. A UK report based on SARS-CoV-2 sequencing among international travelers showed severe cross-border transmission [6] of high-risk variants in this nation. Another report on phylogenetic analysis showed that the SARS-CoV-2 novel variant 20E (EU1) spread immediately to other European countries in the summer of 2020 after its first identification in Spain [7]. Such a rapid spread of EU1 suggests that European travel guidelines and restrictions were primarily inadequate to reduce the risk of cross-border transmission. In addition, it was also estimated that Belgium, the Netherlands, and Norway had more imported cases than exported cases in the summer of 2020, which highlighted the role of imported cases in COVID-19 outbreaks in these countries. In Switzerland, COVID-19 cases associated with cross-border transmission had a considerable impact on the spread dynamics of the local epidemic, which could account for the steady increase of the epidemic in the summers of 2020 and 2021 [8]. Therefore, the risk of COVID-19 transmission between countries cannot be ignored.

To predict the epidemic situation of COVID-19 in a timely, accurate, and reliable manner, scholars have conducted numerous studies on the prediction, prevention and control of COVID-19 transmission [9–13], and an infectious disease dynamics model has been proposed. As a tool aimed at epidemic prediction and as well as actual application, this model considers the transmission speed, transmission mode, and various prevention and control measures of infectious diseases as well as other factors as a whole [14], and thus has significant application value for early warning of infectious diseases as well as for assessing prevention and control effects on the diseases. Reviewing a large amount of literature [15–17], based on the information released by the Chinese Health Commission every day and Hesheng's modeling analysis and prediction of the Wuhan epidemic control and free transmission stages [18], we believe that the cross-border transmission of COVID-19 is divided into two Staged modeling and analysis is currently the best research solution. In the first phase, the epidemic infection status of the exporting country before the entry of imported cases from abroad was simulated and predicted. Hence, the classic SIR model and the random coefficient method were used to calculate the daily existing infected person series and the SIR fitting curve. Then, the SIR model fitting value and the actual value were compared to verify the effectiveness of the model. The other phase included the prediction and analysis of the number of imported cases after the entry of overseas personnel. Specifically, multiple linear regression models were employed to characterize and analyze the impact of the epidemic infection status, the total number of people entering the country, and the effectiveness of prevention and control measures at the airport on the imported cases. The model proposed in this study was used to predict the imported case data in the first half of 2022, which in turn tested the applicability of the model.

## Materials and methods

### SIR model of epidemic infection status in exporting countries before entry

COVID-19 is a particularly contagious respiratory infectious disease, and its pathogen SARA-CoV-2 poses a threat to human life safety, with high transmission efficiency and serious

consequences of infection, despite a low mortality [19]. The incubation period of COVID-19 is generally 7 days and victims will not be reinfected shortly after cure.

Based on this information, we made the following assumptions: (1) The information of the overseas imported patients reported by the health committees of various provinces and cities in China and the information on the epidemic situation of the exporting countries provided by the WHO were true and credible; (2) Only the existing infected population was predicted and analyzed, without potential and suspected patients being considered, and once the confirmed patient came into contact with the susceptible population, it presented a certain degree of infectivity; (3) COVID-19 recoverers would not be reinfected and were not infectious any longer; (4) the natural births and natural death of the population during the transmission of COVID-19 were not considered; and (5) All patients could recover within 14 days after diagnosis and were no longer infectious.

Symbols involved in the model proposed in this study are described as follows: (1) S (susceptible) represents the susceptible population; (2) I (infectious) represents the infected population; and (3) R (recover) represents the recovered population, which are those who are cured. At any time t, the total population N is expressed as N (t) = S (t) + I (t) + R (t). β denotes the probability that the susceptible population S is infected by the infected person I, and γ denotes the probability that the infected person I returns to R.

## Establishment of differential equations based on the classic SIR model

SIR model was first proposed in 1926 and is mainly used to study the epidemic patterns of the black death and plague in London and Mumbai [20]. This model is classical kinetic transmission models that can be used for transmission studies not only of diseases [21] but for of information and computer viruses as well [22, 23]. Drawing on the classical SIR model for infectious disease, We express the spread process of COVID-19 with the following differential equation:

$$\frac{\mathrm{dS}}{\mathrm{dt}} = -\beta SI \tag{1}$$

$$\frac{\mathrm{dI}}{\mathrm{dt}} = \beta SI - \gamma I \tag{2}$$

$$\frac{\mathrm{dR}}{\mathrm{dt}} = \gamma I \tag{3}$$

The propagation process is shown in Fig 1:

## Data sources

**Cases imported from overseas.** Collect information on the entry time, place of entry, time of diagnosis, flight taken, country of export and other information of overseas imported

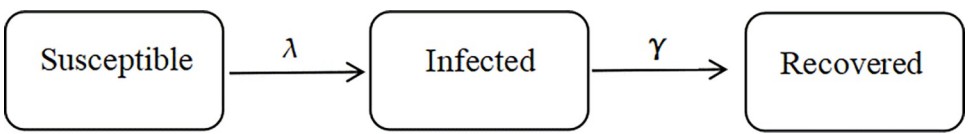

**Fig 1. SIR model infection process.**

cases announced by the health committees of 13 provinces (municipalities) including Beijing and Shanghai in China.

**Flight type and number of passengers taken.** Find the passenger capacity and number of arrivals of the corresponding aircraft model through the port of entry information system.

**The daily cumulative number of confirmed cases in the exporting country.** Under the World Health Organization (WHO) COVID-19 topic (https://www.who.int/emergencies/diseases/novel-coronavirus-2019), find the global data on confirmed cases of new coronavirus infection and select the exporting country with confirmed cases The daily cumulative number of confirmed cases.

**Total country population.** Log in to the Our Word in Data (Our World, https://ourworldindata.org/) data website to collect the latest published total population of each country.

**COVID-19 vaccination status.** Log in to the Our Word in Data website to collect the number and percentage of daily vaccinations in each country.

**Epidemic prevention measures before flight in exporting countries.** When considering the classification of entry policy prevention and control levels, the levels of epidemic prevention measures at the pre-entry airport are uniformly divided according to the requirements for nucleic acid testing, IgG or IgM antibody testing, and antigen testing.

**The prevalence of COVID-19 virus variants at different stages in China's shipping countries.** Collect information on mutant strains through the Our World in Data website.

## Data processing

**Dataset classification.** Data on the imported cases from 51 countries were collected. In the process of SIR model fitting, we divided the dataset as follows: Firstly, the original data regarding the cumulative number of the confirmed cases in the 51 countries from March 1, 2020 to December 31, 2021 were summarized and sorted into time series data; then, the total number of people entering each of the 51 country, the number of confirmed people entering the country, the vaccination rate of the exporting country, the infection rates of variant strains and the epidemic prevention measures taken at the airport were sorted according to the time series, and the obtained data is sorted according to the time series to form a training set. The data from January 1, 2022 to June 30, 2022 composed test data, which were organized as was described above.

**Data splicing and preprocessing.** The original data provide sequential data of the cumulative confirmed numbers in 2020, 2021, and the first half of 2022, respectively, and a time series I (t) of the daily number of existing infections in each country during the epidemic needs to be generated on the basis of the original data.

First, perform an inner connection on the data in 2020 and 2021. The primary key is country, and obtain a breadth table of the historical number of infected people in 51 countries in the past two years. The numerical matrix is recorded as Hi.j{i = 20200824, j = ALB),and the row index i represents time (such as 20200824, 20200825,. . .20210923), the column index j represents the country, and the historical infection number sequence H. j in a single country is recorded as H(t). Then the data H(t) is preprocessed to obtain the time series I(t) of the daily number of infected people in each country. The conversion formula for preprocessing the historical cumulative confirmed number H (t) is as follows:

First, given the historical cumulative number of the confirmed cases per day H (t), R (t) represents the current historical number of recoveries, and because the SIR model does not consider birth and mortality rates during disease transmission, we derived the following equation:

$$H(t) = I(t) + R(t) \tag{4}$$

Second, although we used R (t) to represent the current historical number of recoveries, this part of the data for R (t) is missing, and therefore, we assumed that infected patients can recover within 14 days after the infection. Therefore, Eq (5) is obtained:

$$R(t) - R(t - 14) \approx I(t - 14) \tag{5}$$

Finally, the historical cumulative number of the confirmed cases H (t) 14 days earlier is subtracted from the historical cumulative number of the confirmed cases H (t-14), as in Eq (6):

$$H(t) - H(t - 14) = I(t) + \{R(t - 14) + I(t - 14)\} - \{I(t - 14) + R(t - 14)\} = I(t) \tag{6}$$

Therefore, the resulting sequence for the number of people living with infection I (t) can be expressed as Eq 7:

$$I(t) = H(t) - H(t - 14) \tag{7}$$

**Data screening and missing value handling.** The Data screening and missing data handling steps are as follows;

Step1: Countries with length($\{I(t) \neq 0\}$) $\leq 365$ are excluded, i.e., countries with a sequence I (t) length less than or equal to 365 for non-zero existing infections were excluded because the sequences of these countries are too short for a valid analysis.

Step2: culled $\min(I(t)) \geq 10000000$, $\max(I(t)) \leq 1000$. That is, the sequence of the number of people with existing infections whose minimum value is too large or the maximum value is too small is excluded. Through this step, the national sequence suitable for following analyses can then be basically selected.

Step3: Head data for all 0 in sequence I (t) of existing infections by country is removed.

Step4: Tail data for all 0 in sequence I (t) of existing infections by country is removed.

Deleted data are those which cannot be analyzed or are of minor significance. Data with too short non-zero sequences were removed in the first step because they were difficult to fit in to the SIR model. In the second step, the removal of the sequences whose minimum values were too large or whose maximum value were too small because such data were difficult to account for a very small proportion and thus, they were little meaningful for subsequent analyses. The third and fourth steps eliminated the data with the head and tail of 0, because these data could not fit the SIR model, and their removals would not affect the subsequent analysis.

After the above mentioned steps, the cumulative number of the confirmed cases H (t) in the past three years were collected on the WHO website, and the available sequence I (t) of the existing number of the infected cases were obtained after the cumulative number of the cases H (t-14) confirmed 14 days before was subtracted.

Most of the infection processes of epidemics are complex, and sometimes multiple infection processes may take place in the same time period, or errors may occur in the data collection process. These phenomena can be manifested in complex situations such as abnormal fluctuations in the number of the infected people, as shown in the Fig 2 t2-t3 interval. In our study, after the existing infection sequence I(t) was presented in the form of a curve plot, the existing infection sequence I(t) in each country was screened and waveform-divided. The final selection could be simulated through the SIR model propagation process, and the curve part with a satisfactory fitting effect was simulated and then calculated.

As shown in Fig 2, (t1, t2) and (t3, t4) are the well-fitted portions of the curve for the selected available SIR propagation process. The dates corresponding to t1 and t3 are the starting dates of the intervals, and the dates corresponding to t2 and t4 are the termination dates,

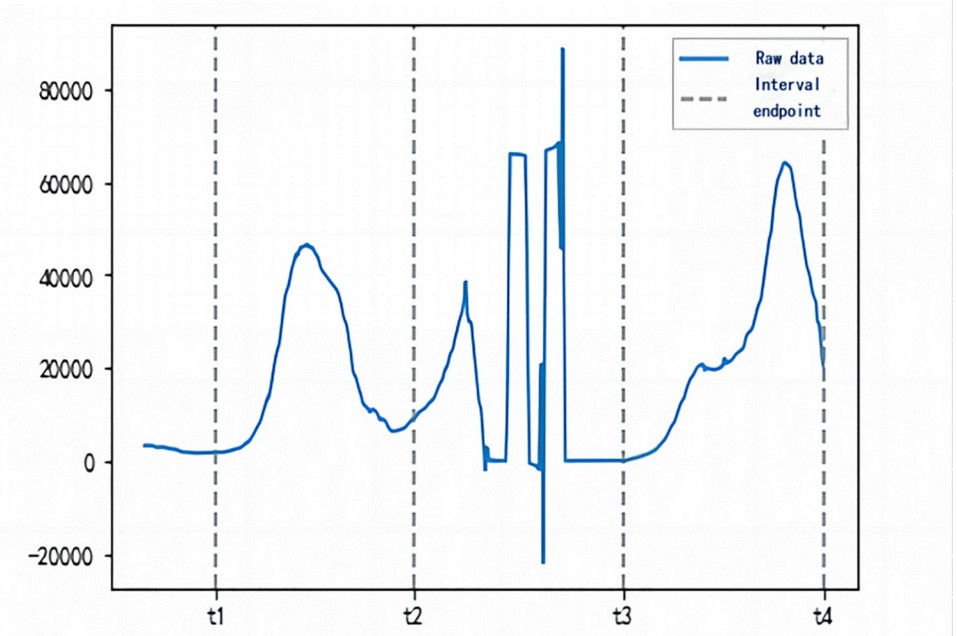

**Fig 2. One of the schematic diagrams of the I(t) peak interval of MMR.**

respectively. The ordinate shows the number of existing infections. The (t2, t3) interval is an abnormal fluctuation that needs to be eliminated.

**Peak pre-segmentation during infection.** During SIR fitting, the peak of the number of the infected people can convey the information on the beginning or the end of an infection process. Therefore, it is an appropriate way to divide the infection process by detecting the peak value. The specific steps are as follows:

Step1: Search the local maximum point against the infection curve I (t) to obtain the point set P of all peaks;

Step2: Set a maximum threshold and a minimum threshold for the peak value, screen the point set P and obtain the point set $p_0$ for all peak values;

Step3; For all points in a certain peak point set $p_0$, set the maximum threshold for a forward detection as well as a backward detection and the threshold for the beginning and end of the propagation, perform a forward detection as well as a backward detection to find the corresponding segmentation points;

Step4: Segment the curve I(t) for the corresponding segmentation points, and the sequence segments too short to be fitted are summarized and filtered.

To seek local maximum points, the following approach is taken:

Step1: Calculate the first order difference of the curve I (t) to obtain diff (I);

Step2: Search for diff (I) to increase the descending mode, and find the mode fragment piece that increases followed by a decrease;

Step3: Take the maximum of the piece as the local maximum.

The searching method for local maximum points can refer to the findPeaks function method in R language.

## SIR model building

In our study, the SIR model was constructed with reference to Florence D´ebarre [24], using the (deSolve) package in R language, and the SIR model presented a good interpretability for

the current spread of the epidemic. The construction of the model required four parameters, including S (susceptible population), β (infection rate), γ (recovery rate), and I (initial number of infections). For the susceptible population S, we adopted a complete infection waveform, and the range of 10-fold amplification of the maximum number of the infected represented the susceptible number. The initial number of infections I was represented by the number of the existing infections I (t), obtained from data processing. Parameters β and γ were automatically generated by the R language.

The SIR model's parameters were obtained by following the stochastic coefficient method [25], and the parameters present in the SIR epidemic model were regarded as random variables with specific distributions. A system of differential equations was obtained based on the stochastic spectral representations of the parameters, and their numerical integration was performed to obtain the corresponding parameters.

In order to efficiently search for the optimization vector and improve the fitting effect of the SIR model, this study uses different parameter combinations as the initial value input to perform traversal search for the optimal fitting effect. N groups are randomly selected from the determined parameter initial value vector as the initial value, and the optimization fitting process is carried out.

The optimization process involved the use of the optim function in R language, and the objective function was the residual sum between the infected number sequence of the fitted SIR model and the actual infected number sequence (calculated based on the dist function in R language). The parameter combination constructed by the SIR model was used as the initial value input, with L-BFGS-B as the optimization method, and the parameters S, β, γ were optimized for output the optimal parameters. For each data, there are N initial values of the optimization process, and the least residuals of the optimization parameters were taken as the outputs. The entire optimization process can be described by the following equation:

$$min_{S,\beta,\gamma,I} \parallel I_{SIR}(S,\beta,\gamma,I) - I_{real} \parallel_2$$

$$s.t. \begin{cases} S > 0 \\ 0 \leq \beta \leq 1 \\ 0 \leq \gamma \leq 1 \\ I > 0 \end{cases} \tag{8}$$

Where $I_{SIR}$ (S, β, γ, I) is the infection sequence of the SIR model determined by the parameters, $I_{real}$ is the real infection sequence, and the optimized objective function is the residual of both (i.e., the L2 norm of the difference). As debugging 25 groups can increase the stability of optimization results, according to the empirical value, we decided on N = 25 to find the optimization results with the smallest residual error.

A schematic diagram of the parameter optimization process for the entire SIR model is shown in Fig 3:

During SIR model fitting, it is necessary to analyze the relevant calculation accuracy and expand the constant term if the fitting is not effective. In the process of parameter fitting, parameter b (β) needs to be multiplied by of 10000, and other parameters need to be extended less. After optimization, the optimal S, β, γ, I of each interval and the coordinates of the peaks can finally be output, as shown in Fig 4:

The solid red line in Fig 4 represents the fitted curve, the dots represent the intercepted original interval, the horizontal axis is time, and the vertical axis is the number of the infected persons. In the process of fitting prediction, it is necessary to record the inaccurate fitting

## Initial value determination and parameter optimization

Country Epidemic Data → Identify intervals and peaks of the infection → Generate objective function

Poor fit

Calculated residuals, optimized parameters ← Generate parameter initial value vector (S,I,b,g)

Ideal fit

Record optimal parameters, output results

**Fig 3. Schematic diagram of optimizing the SIR model parameters.**

curve in the current SIR fitting process during the application process and to add the emerging curve with prediction errors to the configuration table timely. In the meantime, the SIR curve that can be predicted but is not accurately predicted should be refitted or labeled for next step of analysis.

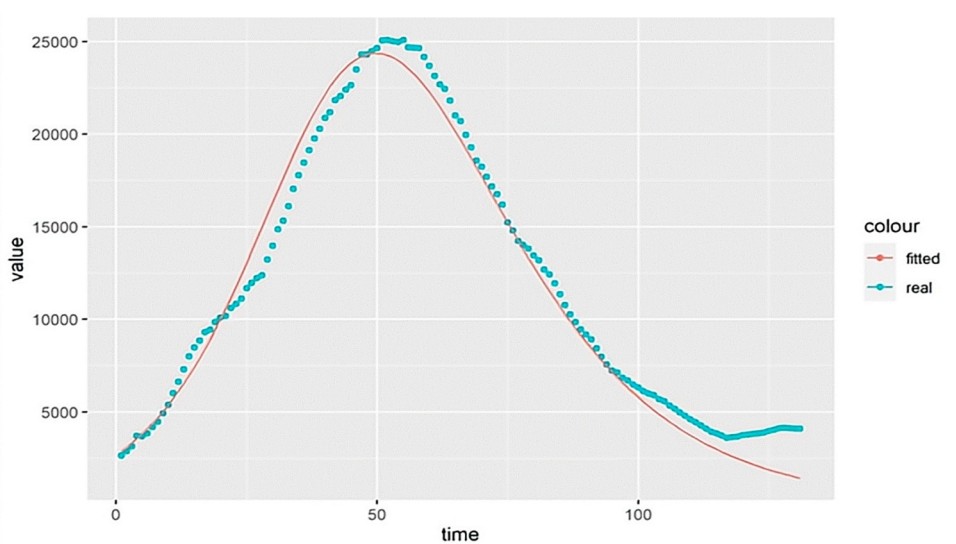

**Fig 4. Schematic diagram of the parameter-optimized fitting results.**

## Multiple linear regression model for predicting the number of imported cases after entry

After SIR model fitting, the epidemic data of each country were obtained. The information contained in the dataset included the number of the existing infections in the exporting country, the number of the inbound persons in the exporting country, the number of the imported cases overseas, the period of the epidemic infection in the exporting country, the country's name and the corresponding time series. Data of the vaccination coverage rate, strain infection efficiency, and epidemic prevention policies were summarized for the corresponding country according to the time series. This information was brought into the multiple linear regression model to construct a prediction model.

The training process of the multiple linear regression model involved the random sampling method, which divided the characteristic data sets of each country obtained from the SIR model fitting according to a training set: test set ratio of 7:3. Multiple linear regression model training and testing were carried out, and appropriate result outputs were selected.

## Combined model prediction process

To describe the import risk more directly, the overall prediction process is described as follows:

Predicted target: The number of the cases imported from overseas within a time period T, case Num;

Predictor variables: Descriptive variable inject Feature List for the infection status of a country within a time period T;

Descriptive variable stage List for the epidemic stage of the country within T;

Descriptive variable total Feature List for the number of people imported outside the country during T.

From a viewpoint of practical application, two prediction processes could be generated in our study, due to the attributes of the SIR model itself, which determined the overall parameter process by some parameters.

**Predicted process 1.** Variables similar to the multiple linear regression prediction model were constructed to predict the linear regression model directly, and the specific process is as follows:

Step1: Identify prediction formula as follows

$$\text{case}_t = \text{pre}\left(inject_t, input_t, stage_t\right) \tag{9}$$

Step2: Construct the following variables:

$Inject_t$: Summary of the number of the domestic infections in the exporting country during the same time period;

$Input_t$: Summary of the number of people imported from abroad by flight during the same time period;

$Stage_t$ (Predicted epidemic Stage): If there are enough observed data (which can be judged according to the original data), the epidemic stage can be defined, it is brought directly into the multiple linear regression prediction model for a prediction; if there are insufficient observed data and the waveform of infection is incomplete, the SIR model can be fitted first to determine the import epidemic stage (as was mentioned in the prediction process 2), which is then brought into multiple linear regression for a prediction.

Step3: predict the number of the infected people casest, i.e., the import risk, on the basis of the parts of the multiple linear regression model.

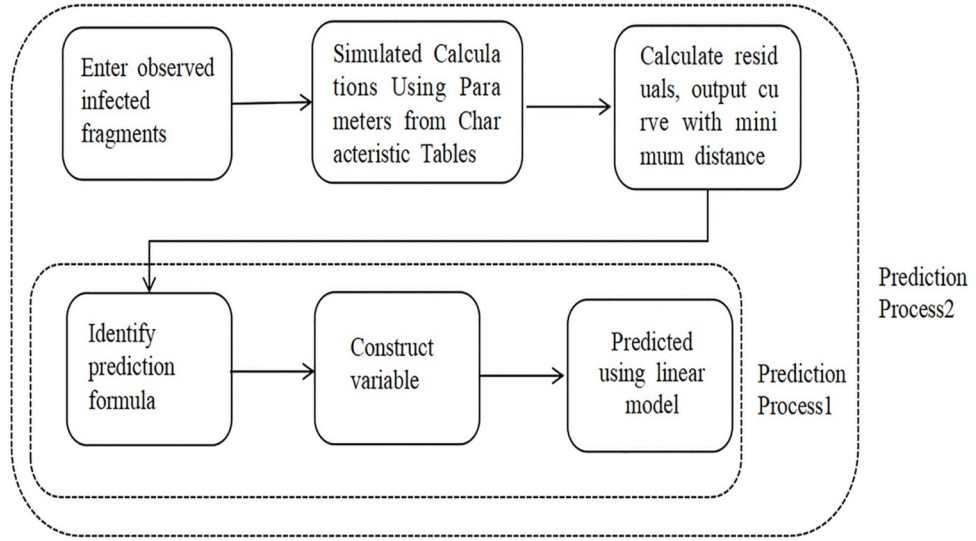

**Fig 5. Schematic diagram of the prediction process.**

**Predictive process 2.** The feature table (the initial value table of SIR fitting) was combined to make prediction, and the specific prediction process is as follows:

Step1: Identify the initial infection segment Inject [t, T] imported overseas, where [t, T] is the observed interval (i.e., the adjacent time period during which the risk of imported infection will be predicted);

Step2: Import the data corresponding to the observation interval into the existing program of R language-based SIR model modeling, and use the relevant SIR model parameters already in the feature table (Table 2) to simulate and calculate the curve I(t);

Step3: In the well-fitted SIR curves, search the part I (t, T) that is close to the curve I (t) obtained from the fitting in Step2, calculate its distance to the initial infection fragment Inject [t, T], calculate the residual error and output the curve with the smallest distance;

Step4: With the curve with the smallest distance from the initial infection fragment Inject [t, T], summarize the features to obtain a dataset similar to the feature table (Table 2). Perform the prediction process 1 and output the predicted value of the number of the imported infections overseas.

A schematic diagram of the prediction process is shown in Fig 5:

## Results

### SIR fitting results for 51 countries is shown in Figs 6–9

### SIR model fitting effect assessment

With the parameter solution method, the optimal parameters for the SIR model were obtained, and these parameters were then used to predict the changes of the epidemic development in the country. The fitting effect of the curve was evaluated by $R^2$, which is formulated as follows:

$$R^2 = 1 - \frac{\sum (Y_i - \hat{Y}_i)^2}{\sum (Y_i - \bar{Y})^2} \tag{10}$$

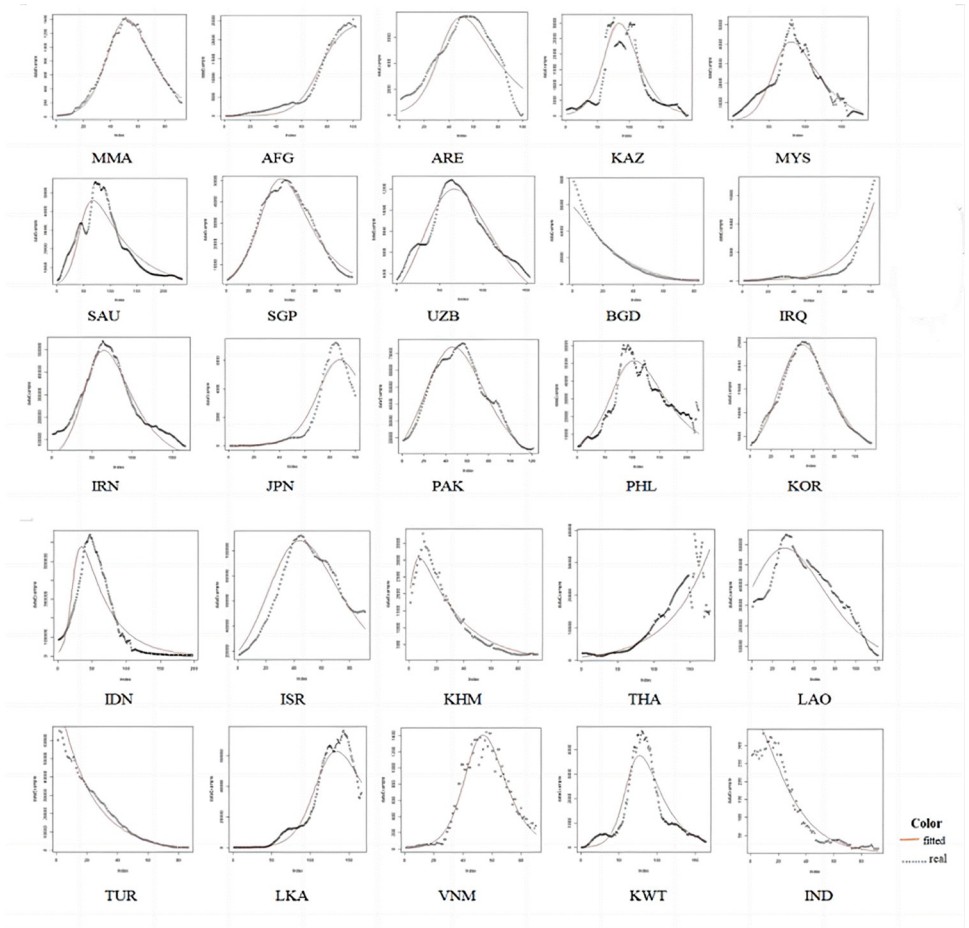

**Fig 6. Comparison of the actual infected persons among different Asian countries based on SIR fitting results.**

where $Y_i$, $\hat{Y}i$, $\bar{Y}$ represents the mean value of each true value, the predicted value, and the sequence, respectively. The closer a $R^2$ value is to 1, the better the curve is considered to be fitted.

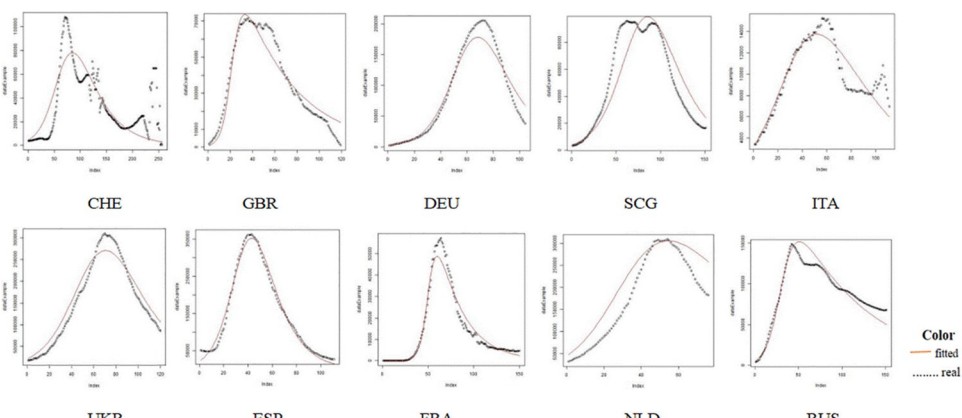

**Fig 7. Comparison of the actual infected persons among different European countries based on SIR fitting results.**

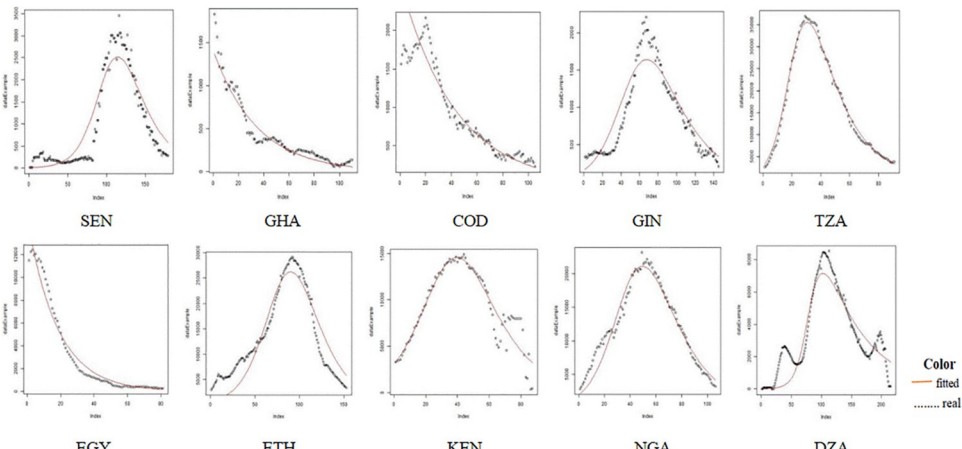

**Fig 8. Comparison of the actual infected persons among different African countries based on SIR fitting results.**

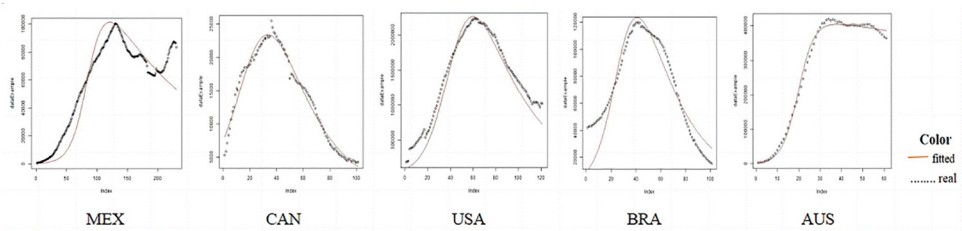

**Fig 9. Comparison of the actual infected persons among different Americas and Oceania countries based on SIR fitting results.**

To test the accuracy of the model, $R^2$ was calculated for each fitted epidemic and a density plot was drawn as shown in Fig 10:

As shown in Fig 10, most of the infection curves in 51 countries have $R^2$ above 0.75, while only a few have $R^2 < 0.4$. Overall, the mean value of all $R^2$ is about 0.86, and this result can indicate that the infection status of COVID-19 cases in different exporting countries before entering China can be well described by the SIR model.

## Multiple regression analysis of the SIR model outputs

After preliminary data processing and SIR fitting, the datasets that were finally entered into the multiple linear regression model were summarized. An explanation table of the variables in this dataset and some examples of the sample are shown in Tables 1 and 2:

## Construction results of the multiple linear regression model

After the feature data set (Table 2) was trained and output from the SIR model, multiple linear regression model results were obtained, as summarized in Table 3:

From Table 3, it can be seen that the number of the infections in the exporting country (injectFeatureList) and the total number of the exporting countries (totalFeatureList) exerted statistically significant effects on the number of the imported cases overseas (P < 0.05). In the whole model, the order of the influencing factors on overseas imported cases was "total

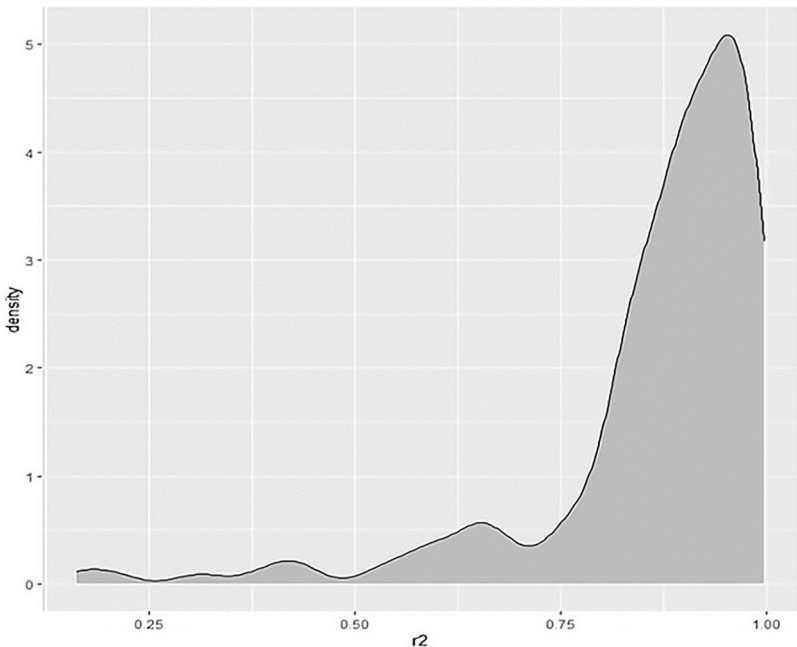

**Fig 10. $R^2$ density plot of SIR model goodness of fit for all infection curves in 51 countries.**

number of the imported cases from exporting countries", "number of the infected cases from exporting countries" and "epidemic stage of the exporting country", from strong to weak.

Finally, the resulting model structure was trained as follows:

$$y = 0.5591 + 1.999 \times 10^{-7} x_{injectFeatureList} + 6.028 \times 10^{-3} x_{totalFeatureList} - 1.012 \times 10^{-1} x_{stageList} + \epsilon \quad (11)$$

From the above formula, it can be obtained that the parameters were positively correlated except for StageList. StageList can be simply understood as follows: The greater the number of the infected people in a country, the greater the total number of the people imported overseas, and the more advanced the stage of epidemic, the higher risk of importation. A slight negative

**Table 1. Interpretation table of the sample data for regression analysis.**

| Variable Name | Variable interpretation |
|---|---|
| InjectFeatureList | Sequence of the number of people infected in the exporting country |
| Country | Output country abbreviation |
| PeaksListVec | Peak sequence |
| StageList | Epidemic stage of the exporting country |
| TotalFeatureList | Total entered |
| CaseFeatureList | Number of the cases entered |
| sVecOpt | Parameter of the susceptible number of the transporting country |
| bVecOpt | Parameter of the infection rate of the transporting country |
| gVecOpt | Parameter of the transportation recovery rate |
| initialOpt | Initial number of the people infected in the current epidemic in the exporting country |
| timeColSeries | Time series of the infections of the case exporting country |
| vaccinateMaxVec | Maximum vaccination rate of the case exporting countries at the current stage |
| virusFeatureItem | Virus infection rate of the case exporting country |
| policyLevel | Entry policy average |

**Table 2. Examples of the regression analysis sample part for SIR model fitting.**

| InjectFeatureList | Country | Peaks ListVec | StageList | Total FeatureList | Case FeatureList | SVecOpt | BVecOpt | GVecOpt | InitialOpt |
|---|---|---|---|---|---|---|---|---|---|
| 5225362 | USA | Peaks1 | 3 | 1651 | 9 | 421657 | 0.006287 | 0.00227 | 2941 |
| 541663 | RUS | Peaks1 | 3 | 0 | 0 | 155557 | 0.009496 | 0.07744 | 3154 |
| 349247 | AFG | Peaks1 | 2 | 0 | 0 | 24737 | 0.043451 | 0.00000 | 10 |
| 1531344 | MMR | Peaks1 | 3 | 630 | 1 | 203779 | 0.001036 | 0.00000 | 1946 |
| 2241166 | CAN | Peaks2 | 2 | 784 | 9 | 585636 | 0.001563 | 0.04304 | 1387 |
| 2153999 | GBR | Peaks1 | 4 | 0 | 1 | 568783 | 0.001687 | 0.04733 | 3881 |
| 2640332 | ESP | Peaks3 | 2 | 1046 | 4 | 693105 | 0.002743 | 0.03162 | 27986 |
| 385060 | ETH | Peaks4 | 4 | 1574 | 15 | 10623 | 0.073584 | 0.00557 | 4395 |
| 1235627 | FRA | Peaks3 | 4 | 2145 | 9 | 681777 | 0.001179 | 0.01309 | 36529 |
| 630640 | IRQ | Peaks4 | 0 | 546 | 1 | 145860 | 0.001944 | 0.00808 | 17506 |

Note: This table only shows some examples for the multiple linear regression model

correlation between the numerical codes (i.e. stageList) assigned to the epidemic stage and the target value may be on account of a further reduction in infectivity later in the epidemic stage.

## Multiple linear regression model residual analysis

Residual error is the difference between the actual value and the predicted value. Residual analysis can be used to detect the rationality of the model assumption and the reliability of the data. It is an effective tool to check the compliance of the data with the model. The distributions of the residuals of the above models for the training set and the test set are shown in Figs 11 and 12, respectively.

As shown in the figures above, the residuals of the test set and the middle part of the residuals of the training set basically met a normal distribution, indicating that the fitting effect of the model is good within these parts. The partial outliers in the training set versus the test set caused the graphs to present partial skewness.

From the Table 4, the test set appeared consistently with the training set at the position of the predicted median, exhibiting relative stability. Within the 75% quantile, the predicted values of both the test and training sets were slightly larger than the actual values (with negative residuals). After the 75% quantile, the predicted values turned gradually smaller than the actual values, and the maximum deviation value of the test set was noticeably greater than the that of the training set.

## Multiple linear regression model validation

The 2022 test set data was used to predict overseas imported cases, and the results were as follows:

**Table 3. Significance test results of the optimized model.**

| Variable | Coefficient | Standard Error | t value | P value |
|---|---|---|---|---|
| Constant term | $5.591 \times 10^{-1}$ | $2.806 \times 10^{-1}$ | 1.993 | 0.047 |
| InjectFeatureList | $1.999 \times 10^{-7}$ | $2.230 \times 10^{-8}$ | 8.963 | < 0.001 |
| StageList | $-1.012 \times 10^{-1}$ | $7.153 \times 10^{-2}$ | -1.415 | 0.1575 |
| TotalFeatureList | $6.028 \times 10^{-3}$ | $1.799 \times 10^{-4}$ | 33.054 | < 0.001 |

Note: This regression model was statistically significant (F = 538.7, P < 0.001, adjusted R2 = 0.7).

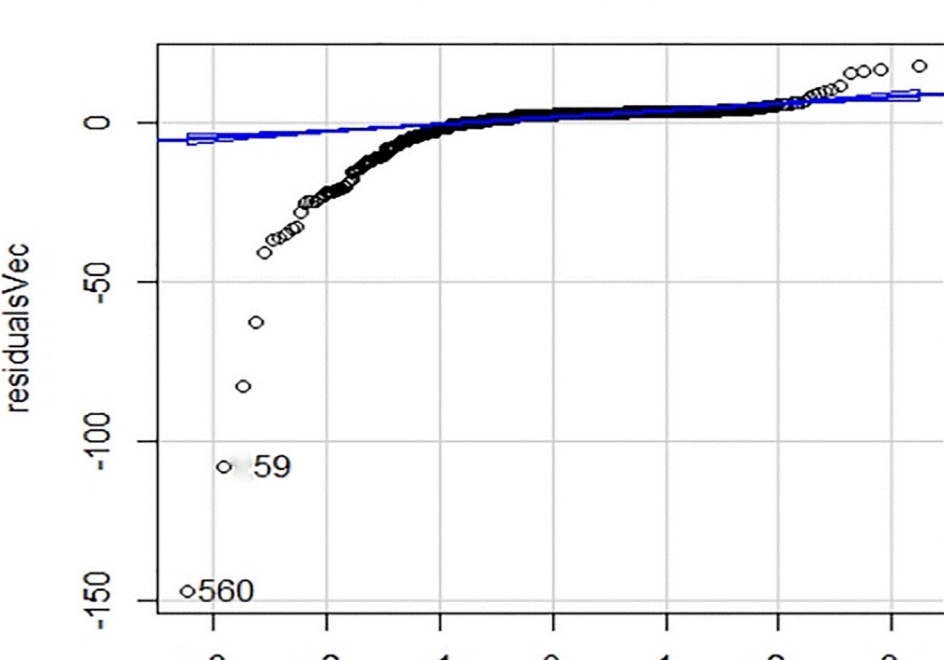

**Fig 11. qq-plot plot of the residuals for the training set.**

As shown in Fig 13, The model we established predicts that the number of overseas imported cases is overall higher than the actual number of overseas imported cases. The reason may be that international flights were still under control when the data were collected and there were not flights arriving every day, which is different from the data collected. It is related to the fact that the total number of immigrants entering the country is 0 at some times, which results in the predicted value being higher than the actual value. At present, flight control has been fully relaxed, and the prediction model established by this research will have better applications in actual predictions in the future.

## Predicted result evaluation

As shown in Fig 14, the residuals of the prediction set met a normal distribution, indicating that the prediction effect of the model on the prediction set in 2022 was excellent.

## Discussion

Since the first outbreak of COVID-19, predicting its development trend has never been stopped, and a number of prediction models have been proposed, such as the combined prediction models [26] based on complete ensemble empirical modal decomposition (CEEM-DAN), extreme gradient elevation tree (XGBoost), and network search data (WSD). Also, methods [27] for predicting the development trend of COVID-19 have been attempted, such as the long- and short-term memory (LSTM) neural network of Dropout technology and the SEIR optimization model [28]. However, CEEMDAN, XGBoost and WSD are complex and

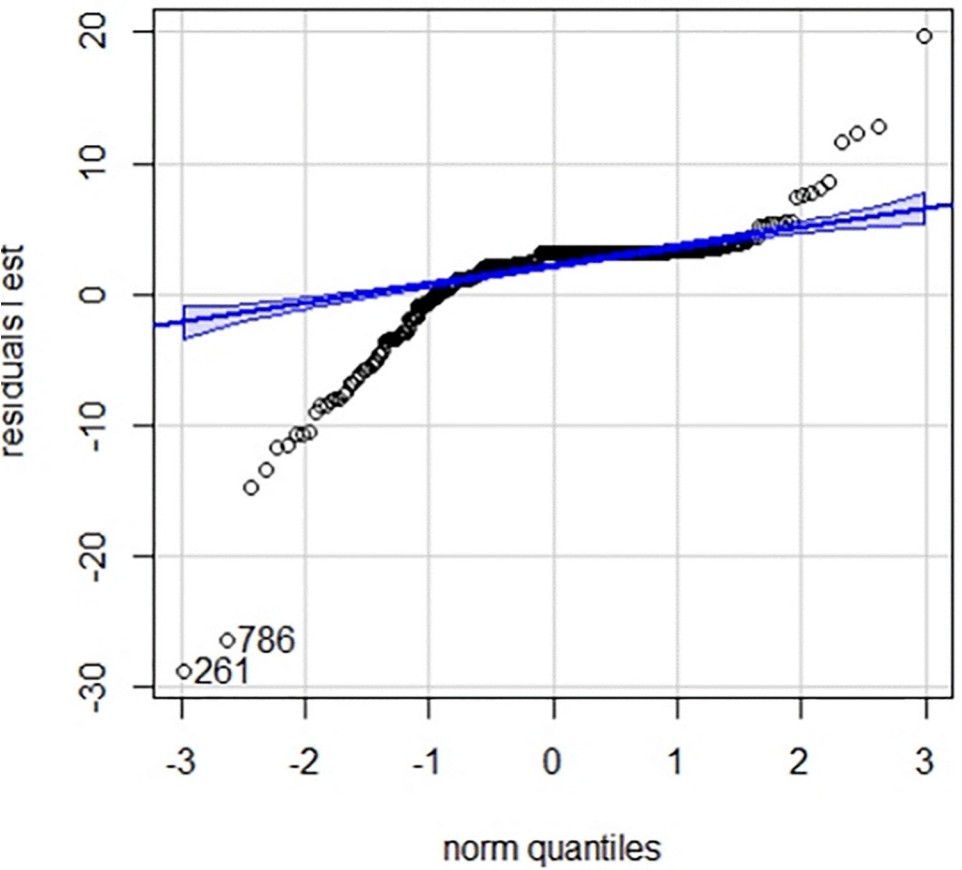

**Fig 12. qq-plot plot of the residuals for the test set.**

require high timeliness of the data [26]. For the LSTM neural network approach to predict the development trend of COVID-19 [27], the error interval is too large when it is used to predict the cumulative confirmed numbers based on a large base number within a long statistical time. Although the SEIR model considers the impact of novel corona virus infection latency on the development of the epidemic, the latency individuals in the actual transmission process, the difference in the onset time is difficult to count [29], and objective factors such as isolation measures have a greater impact on it, resulting in that the prediction model cannot accurately obtain the real epidemic parameters [28]. In the actual spread of the epidemic, it is reasonable to assume that the infection development and change of the infected persons and latent persons in a country can be finally reflected in the total number of the confirmed persons in that country, based on which a better development-predicting outcome can be obtained, this

**Table 4. Comparison of the residuals between the test set and the training set (real-predict).**

| Collections | Min | 1st | Median | Mean | 3rd | Max |
|---|---|---|---|---|---|---|
| Train | -17.7192 | -3.5400 | -2.3777 | 0.0000 | -0.3776 | 148.9389 |
| Test | -17.2856 | -3.5621 | -2, 5841 | 0.6553 | -0.0184 | 340.0221 |

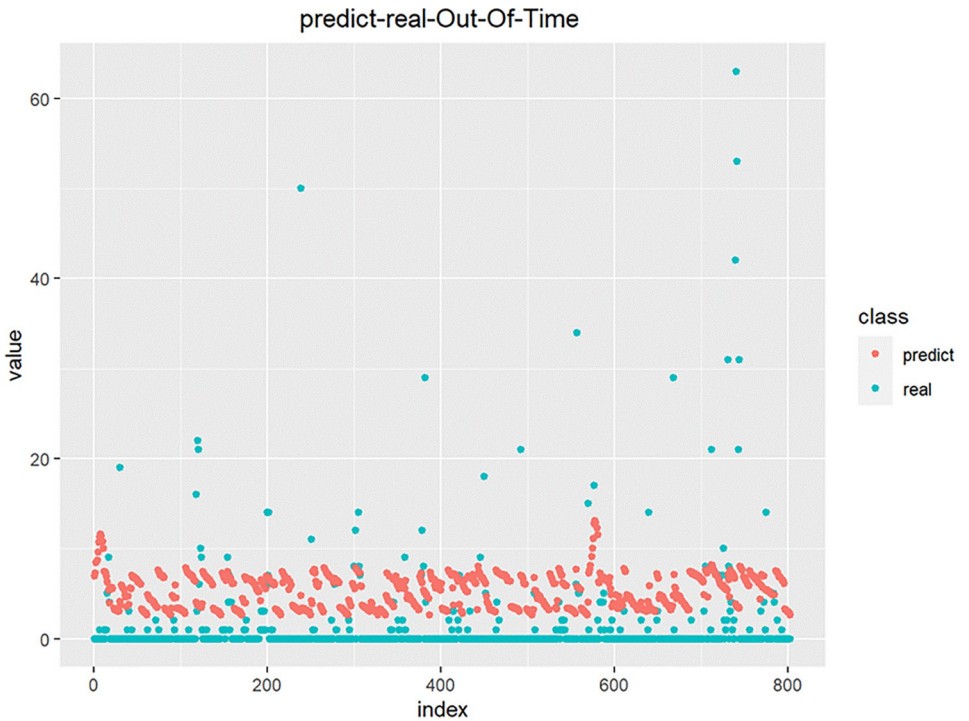

**Fig 13. SIR-multiple linear regression model 2022 forecast set data forecast results chart.**

assumption gets support from Kremer [30]. Therefore, the most classical SIR model of infectious diseases was finally selected in our study for epidemic evolution prediction. Also, according to the existing studies, the direct use of the SIR model [31] has been proved effective in

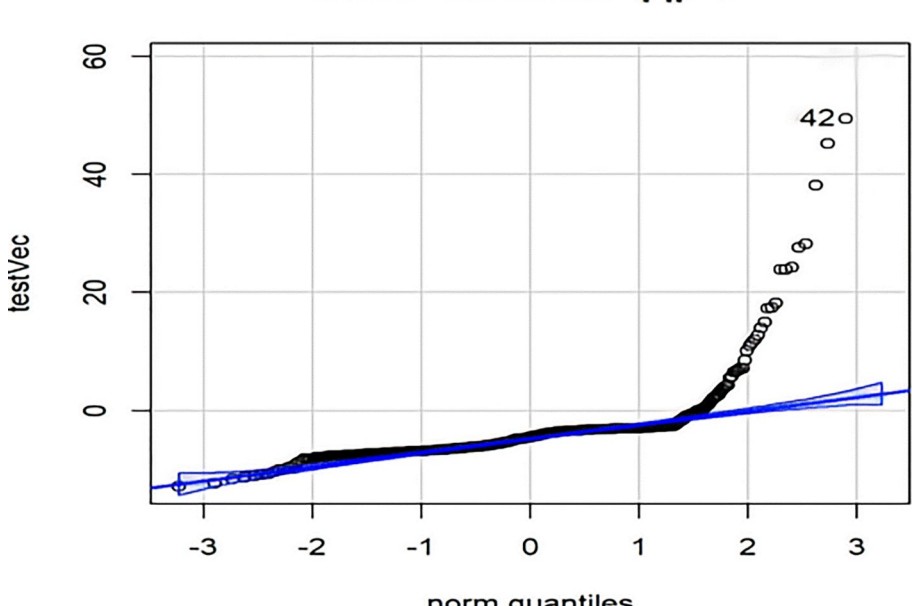

**Fig 14. qq-plot of the predicted set residuals in 2022.**

performing such prediction tasks with good interpretability for the current spread of the epidemic.

For the final overseas imported cases prediction part, the multiple linear regression model used in our study seemed slightly weaker compared with other prediction models. However, Smita Rath used multiple linear regression model to predict the epidemic situation of COVID-19 in India. To do that, he assessed the epidemic data in India by reviewing the historical applications of the model and found that this model could achieve good prediction results [32]. His findings indicates that multiple linear regression could be used as a satisfactory tool for predicting COVID-19. Similarly, Hari predicted the number of deaths caused by COVID-19 pneumonia using a regression model based on the data collected at the Hopkins Data website [33]. Multiple linear regression prediction models have high predictive accuracy, and Bakhtiarvand's regression model used to analyze and predict the severity of COVID-19 patients evidenced for this point [34]. Considering that the data used in our study are linear, the multiple linear regression model should be more suitable for our prediction. The results of the comprehensive significance test showed that the order of the influencing factors on the number of the imported cases abroad, from strong to weak, was as follows: the number of infections in the exporting country, the number of the imports from the exporting country, and the period of the epidemic in the exporting country. Although the multiple linear regression model established in our study led to slightly higher prediction values, which might be related to the fact that during the data collection period, China 's international flights were under the management and control according to the "Five Ones" policy (one airline company retains one flight route in one country and up to one flight in one week) [35, 36]. In this study, the multiple linear regression model was utilized to predict the number of inbound persons within a period of time. The restriction upon the number of flights during the management and control period might partially contribute to the higher prediction results in this study. Furthermore, the data collected from the WHO website on the COVID-19 epidemic in the exporting countries may suffer a time lag between epidemiological discovery and data uploading, which might also affect the accuracy of the prediction results to some extent. Neverthelss, with the gradual removal of the national epidemic prevention and control policies of China, the cross-border transmission prediction model established in our study can be further tested and verified with larger data same sizes.

## Conclusions

In our study, the epidemic infection status of the imported cases before entering the country and the cross-border transmission process of COVID-19 after entering the country were modeled and analyzed. The SIR model could well predict the infection status of the overseas epidemic situation by constructing the overseas import combined prediction model based on the overseas import case data. The SIR-multiple linear regression combination model has a good overall prediction effect when used to predict the risk of imported cases from abroad.

### Limitations and future recommendation

Limitations of our study included the following points. First, the dimension for data collection in this study was relatively simple, and data with regard to other variables that affected the effect of the model fitting (such as the rate of mask wearing in the country, the number of open public places, government regulatory measures, etc.) are difficult to collect. Excluding these factors from modeling might affect the accuracy of the model. Second, the multiple linear regression model in itself suffers from limited fitting capacity. Consequently, the actual fitting effect might be deficient, and as such, the final prediction results of this study might be

affected. Third, the influence of different influencing factors on the prediction results considerably varies, with some factors exerting too weak influence to be reflected in the model in the correlation analysis.

The SIR model was used to fit the epidemic transmission status of a country, and then the domestic epidemic parameters were brought into the multiple linear regression model to predict the number of the imported cases. The SIR-multiple linear regression combined prediction model transformed the magnitude of the cross-border import likelihood into more intuitive and visible values and thus provide a reference for port judgment on the risk of overseas import.The model is used to predict the risk of overseas importation after the liberalization of the epidemic situation in China, and will have a better prediction effect.

## Supporting information

**S1 Dataset. These data are available at WHO coronavirus disease (COVID-19).** This website contains daily epidemic information of overseas imported countries. https://www.who.int/emergencies/diseases/novel-coronavirus-2019.
(CSV)

**S2 Dataset. Summary of datasets used in multiple linear regression models.** This data is used to predict cases imported from abroad.
(XLSX)

**S3 Dataset. Summary of datasets fitted by the SIR models.** This data is used to predict the prevalence of COVID-19 abroad.
(CSV)

**S4 Dataset. Summary of datasets used in multiple linear regression models.** This data is used to train the Multiple Linear Regression Models.
(CSV)

## Author Contributions

**Conceptualization:** Ying Wang, Yueqian Song, Jie Tian, Jianzhou Yang.

**Data curation:** Huaxiang Rao.

**Formal analysis:** Ying Wang, Fang Yuan, Huaxiang Rao, Lili Xiao, Xiaolong Zhang.

**Funding acquisition:** Jie Tian, Jianzhou Yang.

**Investigation:** Ying Wang, Fang Yuan, Mufan Li, Jiayu Wang, Yi zhou Ren.

**Methodology:** Yueqian Song, Lili Xiao, Huilin Guo.

**Project administration:** Jie Tian, Jianzhou Yang.

**Resources:** Jie Tian.

**Software:** Ying Wang, Fang Yuan.

**Supervision:** Yueqian Song, Jie Tian, Jianzhou Yang.

**Validation:** Lili Xiao, Huilin Guo, Xiaolong Zhang.

**Visualization:** Ying Wang, Fang Yuan.

**Writing – original draft:** Ying Wang, Fang Yuan.

**Writing – review & editing:** Huaxiang Rao, Jie Tian, Jianzhou Yang.

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
