## [Decision Letter · Decision Letter 0]

7 Aug 2023

PONE-D-23-14890Prediction of cross-border spread of the COVID-19 pandemic: A predictive model for imported cases outside ChinaPLOS ONE

Dear Dr. Yang,

Thank you for submitting your manuscript to PLOS ONE. After careful consideration, we feel that it has merit but does not fully meet PLOS ONE’s publication criteria as it currently stands. Therefore, we invite you to submit a revised version of the manuscript that addresses the points raised during the review process.

We look forward to receiving your revised manuscript.

Kind regards,

Haris Khurram

Academic Editor

PLOS ONE

“This study was funded by the Research Program of the General Administration of Customs of China (Grant Number, 2021HK147) and the Research and Development Project of Non-contact Health Quarantine Supervision Technology Equipment and Systems of Entry Vehicles (Grant Number 2022HK144), as well as Four “Batches” Innovation Project of Invigorating Medical through Science and Technology of Shanxi Province(Grant Number, 2022XM45).”

Reviewers' comments:

Reviewer's Responses to Questions

**Comments to the Author**

1. Is the manuscript technically sound, and do the data support the conclusions?

Reviewer #1: Yes

Reviewer #2: Yes

2. Has the statistical analysis been performed appropriately and rigorously? 

Reviewer #1: Yes

Reviewer #2: Yes

3. Have the authors made all data underlying the findings in their manuscript fully available?

Reviewer #1: No

Reviewer #2: No

4. Is the manuscript presented in an intelligible fashion and written in standard English?

Reviewer #1: Yes

Reviewer #2: Yes

5. Review Comments to the Author

Reviewer #1: 1. The author should carefully check the Abstract and rewrite it. For instance, the sentence "Most of the curve R2 .....which indicate a satisfactory model fit." is confusing. Additionally, the author claimed that the R2 of the classical SIR model is above 0.75, while the R2 of the new proposed SIR-multiple linear regression combined model is R2=0.7. It seems the proposed model can not beat the classical SIR model. Is that right?

2. The introduction part has many typos. The author should carefully check it. For instance, "an infectious disease dynamic model has been proposed should be "many infectious disease dynamic models have been proposed".

3. The author should clearly claim the contribution of this work. For instance, list the contribution as 1. xxxx, 2xxx, 3. xxxx.

4. The section "Establishment of differential equations based on the classic SIR model" is too long, as the SIR model is well-known to researchers. The author should shrink this part. For instance, Fig. 1 can be removed.

5. In the data processing part, the source of the dataset should be given. Then, the description of the original data should be given.

6. The paragraph "First, the data from 2020 and from 2021 .... is denoted as H(t)" should be revised. It is confusing.

7. The author should present an example of matrix $H_{i,j}$.

8. The I(t) of Figure 2 is the cumulative number of the 51 countries or just one country. A detailed description of Figure 2 should be given.

9. The subsection "peak pre-segmentation during infection" should be presented as an algorithm form.

10. Figure 4 shows the data of which country? And which value? The number of I?

11. New SARS-CoV-2 variants are one of the main factors that bring multiple waves of the pandemic. The author should mention this in the introduction part and cite related works, such as "Zhan C, Tse CK, Fu Y, Lai Z, Zhang H. Modeling and prediction of the 2019 coronavirus disease spreading in China incorporating human migration data. Plos one. 2020 Oct 27;15(10):e0241171." and "Zhan C, Zheng Y, Shao L, Chen G, Zhang H. Modeling the spread dynamics of multiple-variant coronavirus disease under public health interventions: A general framework. Information Sciences. 2023 May 1;628:469-87."

Reviewer #2: Abstract is not completed and it must be included the main findings and effectiveness of the model, as there are already several published research articles based on SIR model.

Some more relevant references must be added to the Introduction, why the proposed modelling approach is more suitable than other methods and compare it. Authors must mention the data collection reference in the data processing section. Figure captions must be self sufficient. Fig. 10 and 13 must be described elaborately. The results section is interesting; however, the proposed method must be described in the conclusion section why it is necessary in the current scenario with other recent published articles. Tables can be shifted to the Appendix section.

6. PLOS authors have the option to publish the peer review history of their article (what does this mean?). If published, this will include your full peer review and any attached files.

Reviewer #1: No

Reviewer #2: No

---

## [Author Response · Author response to Decision Letter 0]

15 Oct 2023

September 19, 2023

Dear Editor, 

Thank you very much for giving us a chance to submit a revised version of the manuscript entitled “Prediction of cross-border spread of the COVID-19 pandemic: A predictive model for imported cases outside China” . We have made changes according to comments of editor and the reviewers. Attached please find the revised manuscript with changes highlighted. Below are our point-to-point response to each revised comment.

We hope our revision and response to each comment would meet the requirement of the editor and the reviewers of Plos one.

Best

Sincerely,

Jianzhou Yang 

Responses to Reviewer #1

Question 1. The author should carefully check the Abstract and rewrite it. For instance, the sentence "Most of the curve R2 .....which indicate a satisfactory model fit." is confusing. Additionally, the author claimed that the R2 of the classical SIR model is above 0.75, while the R2 of the new proposed SIR-multiple linear regression combined model is R2=0.7. It seems the proposed model can not beat the classical SIR model. Is that right?

Reply 1: Thank you very much for your suggestion, we have revised the abstract. We have responded to your questions as follows. First of all, "most of the curve R2...this indicates a satisfactory model fit". This paragraph is used to describe our use of the SIR model to fit the epidemic infection situation in overseas exporting countries. We have analyzed the existing data The SIR model was fitted to the epidemic spread status of 51 countries, and the result was that the R2 of most curves was above 0.75. Secondly, for the question "theR2 of the classic SIR model is higher than 0.75, while the R2 of the newly proposed SIR-multiple linear regression combination model is R2=0.7, the proposed model does not seem to be able to beat the classic SIR model" we have the following explanation , the classic SIR model is only used to simulate the spread of epidemics in overseas countries, and the epidemic data obtained is the epidemic data of overseas exporting countries; while the newly proposed SIR-multiple linear regression combination model includes two processes, and after fitting the overseas export After analyzing the country's epidemic spread situation, the fitted data is used to predict the risk of imported cases from abroad. The two models are applied for different purposes, so we do not agree with the statement that the proposed model does not seem to be able to beat the classic SIR model.

Question 2. The introduction part has many typos. The author should carefully check it. For instance, "an infectious disease dynamic model has been proposed should be "many infectious disease dynamic models have been proposed".

Reply 2: Thank you very much for your suggestion. We have checked the spelling issues in the article.

Question 3. The author should clearly claim the contribution of this work. For instance, list the contribution as 1. xxxx, 2xxx, 3. xxxx.

Reply 3: Thank you very much for your suggestion. The description of the author's contribution in the article is written according to the journal submission requirements, and we have made a clearer classification of the author's contribution.

Question 4. The section "Establishment of differential equations based on the classic SIR model" is too long, as the SIR model is well-known to researchers. The author should shrink this part. For instance, Fig. 1 can be removed.

Reply 4: Thank you very much for your suggestion. We have deleted this part of the content. (see text for details)

Question 5. In the data processing part, the source of the dataset should be given. Then, the description of the original data should be given.

Reply 5: Thank you very much for your suggestion. We have modified the data processing part and added the source of each data.Details are as follows：

“Data Sources 

Cases imported from overseas. Collect information on the entry time, place of entry, time of diagnosis, flight taken, country of export and other information of overseas imported cases announced by the health committees of 13 provinces (municipalities) including Beijing and Shanghai in China.

Flight type and number of passengers taken. Find the passenger capacity and number of arrivals of the corresponding aircraft model through the port of entry information system.

The daily cumulative number of confirmed cases in the exporting country. Under the World Health Organization (WHO) COVID-19 topic (https://www.who.int/emergencies/diseases/novel-coronavirus-2019), find the global data on confirmed cases of new coronavirus infection and select the exporting country with confirmed cases The daily cumulative number of confirmed cases.

Total country population. Log in to the Our Word in Data (Our World, https://ourworldindata.org/) data website to collect the latest published total population of each country.

COVID-19 vaccination status. Log in to the Our Word in Data data website to collect the number and percentage of daily vaccinations in each country.

Epidemic prevention measures before flight in exporting countries. When considering the classification of entry policy prevention and control levels, the levels of epidemic prevention measures at the pre-entry airport are uniformly divided according to the requirements for nucleic acid testing, IgG or IgM antibody testing, and antigen testing.

The prevalence of COVID-19 virus variants at different stages in China’s shipping countries. Collect information on mutant strains through the Our World in Data website.”

Question 6. The paragraph "First, the data from 2020 and from 2021 .... is denoted as H(t)" should be revised. It is confusing.

Reply 6: Thank you very much for your suggestion. We have modified this part. Modify as follows: First, perform an inner connection on the data in 2020 and 2021. The primary key is country, and obtain a breadth table of the historical number of infected people in 51 countries in the past two years. The numerical matrix is recorded as Hi j, and the row index i represents time (such as 20200824, 20200825, ...20210923), the column index j represents the country, and the historical infection number sequence H. j in a single country is recorded as H(t). Then the data H(t) is preprocessed to obtain the time series I(t) of the daily number of infected people in each country.

Question 7. The author should present an example of matrix $H_{i,j}$.

Reply 7: Thank you very much for your suggestion, we have modified it in the text.

“The numerical matrix is recorded as Hi.j{i=20200824, j= ALB),and the row index i represents time (such as 20200824, 20200825, ...20210923), the column index j represents the country, and the historical infection number sequence H. j in a single country is recorded as H(t). ”

Question 8. The I(t) of Figure 2 is the cumulative number of the 51 countries or just one country. A detailed description of Figure 2 should be given.

Reply 8:Thank you very much for your suggestion. We have modified the description of Figure 2.

“Fig 2. One of the schematic diagrams of the I(t) peak interval of MMR”

Question 9. The subsection "peak pre-segmentation during infection" should be presented as an algorithm form.

Reply 9: Thank you very much for your suggestion. We are sorry that the peak pre-segmentation part of the infection process cannot be displayed in the form of an algorithm in the article. This article is currently applying for a patent. Peak pre-segmentation is a core point in the patent. Please forgive us for not being able to display it at the moment. We sincerely apologize to you.

Question 10. Figure 4 shows the data of which country? And which value? The number of I?

Reply 10: Thank you very much for your suggestion. The data shown in Figure 4 is only the data of Myanmar among the 51 countries. We selected one of the many infection curves after optimizing the fitting parameters of the Myanmar SIR model.

Question11. New SARS-CoV-2 variants are one of the main factors that bring multiple waves of the pandemic. The author should mention this in the introduction part and cite related works, such as "Zhan C, Tse CK, Fu Y, Lai Z, Zhang H. Modeling and prediction of the 2019 coronavirus disease spreading in China incorporating human migration data. Plos one. 2020 Oct 27;15(10):e0241171." and "Zhan C, Zheng Y, Shao L, Chen G, Zhang H. Modeling the spread dynamics of multiple-variant coronavirus disease under public health interventions: A general framework. Information Sciences. 2023 May 1;628:469-87."

Reply 11: Thank you very much for your suggestion, we have revised the introduction.“Zhan C predicted the development of the epidemic and concluded that high COVID-19 contagion rates[3] and new SARS-CoV-2 variants are one of the main factors leading to multiple waves of the pandemic[4]. In the face of a highly contagious and highly mutable virus, if a country's public health system lacks preparation to deal with its cross-border spread, or if quarantine management and isolation measures for imported cases are underdeveloped, the risk of COVID-19 spread will Increase[5].”

Responses to Reviewer #2

Abstract is not completed and it must be included the main findings and effectiveness of the model, as there are already several published research articles based on SIR model.

Some more relevant references must be added to the Introduction, why the proposed modelling approach is more suitable than other methods and compare it. Authors must mention the data collection reference in the data processing section. Figure captions must be self sufficient. Fig. 10 and 13 must be described elaborately. The results section is interesting; however, the proposed method must be described in the conclusion section why it is necessary in the current scenario with other recent published articles. Tables can be shifted to the Appendix section.

Question1. Abstract is not completed and it must be included the main findings and effectiveness of the model, as there are already several published research articles based on SIR model.

Reply1: Thank you very much for your suggestion, we have added the model validity content to the abstract.

Question2. Some more relevant references must be added to the Introduction, why the proposed modelling approach is more suitable than other methods and compare it.

Reply2: Thank you very much for your suggestion, we have added the relevant content in the introduction. “Currently, there are relatively few models for predicting overseas imported cases.Reviewing a large amount of literature [15-17], based on the information released by the Chinese Health Commission every day, and Hesheng’s modeling analysis and prediction of the Wuhan epidemic control and free transmission stages [18], we believe that the cross-border transmission of COVID-19 is divided into two Staged modeling and analysis is currently the best research solution.”

Question3. Authors must mention the data collection reference in the data processing section.

Reply3: Thank you very much for your suggestion. We have added information related to data collection in the data processing section. See text for details

Question 4. Figure captions must be self sufficient. Fig. 10 and 13 must be described elaborately. 。

Reply 4: Thank you very much for your suggestion. We have described Figure 10 and Figure 13 in detail.

“As shown in Figure 10, most of the infection curves in 51 countries have R2 above 0.75, while only a few have R2 < 0.4. Overall, the mean value of all R2 is about 0.86, and this result can indicate that the infection status of COVID-19 cases in different exporting countries before entering China can be well described by the SIR model.”

“As shown in Figure 13, The model we established predicts that the number of overseas imported cases is overall higher than the actual number of overseas imported cases. The reason may be that international flights were still under control when the data were collected and there were not flights arriving every day, which is different from the data collected. It is related to the fact that the total number of immigrants entering the country is 0 at some times, which results in the predicted value being higher than the actual value. At present, flight control has been fully relaxed, and the prediction model established by this research will have better applications in actual predictions in the future.”

Question 5. The results section is interesting; however, the proposed method must be described in the conclusion section why it is necessary in the current scenario with other recent published articles.

Reply 5: Thank you very much for your suggestion, we have made changes in the conclusion section.

Question 6. Tables can be shifted to the Appendix section.

Reply 6: Thank you very much for your suggestion. We have added the abbreviations of 51 countries in the appendix, as well as the data in Table 2.

---

## [Decision Letter · Decision Letter 1]

18 Mar 2024

Prediction of cross-border spread of the COVID-19 pandemic: A predictive model for imported cases outside China

PONE-D-23-14890R1

Dear Dr. Yang,

We’re pleased to inform you that your manuscript has been judged scientifically suitable for publication and will be formally accepted for publication once it meets all outstanding technical requirements.

Kind regards,

Nenad Filipovic

Academic Editor

PLOS ONE

Additional Editor Comments (optional):

The manuscript can be accepted now.

Reviewers' comments:

Reviewer's Responses to Questions

**Comments to the Author**

1. If the authors have adequately addressed your comments raised in a previous round of review and you feel that this manuscript is now acceptable for publication, you may indicate that here to bypass the “Comments to the Author” section, enter your conflict of interest statement in the “Confidential to Editor” section, and submit your "Accept" recommendation.

Reviewer #1: All comments have been addressed

2. Is the manuscript technically sound, and do the data support the conclusions?

Reviewer #1: Yes

3. Has the statistical analysis been performed appropriately and rigorously? 

Reviewer #1: Yes

4. Have the authors made all data underlying the findings in their manuscript fully available?

Reviewer #1: Yes

5. Is the manuscript presented in an intelligible fashion and written in standard English?

Reviewer #1: Yes

6. Review Comments to the Author

Reviewer #1: All the questions have been addressed. I have no comments any more. I recommend to accept this paper.

7. PLOS authors have the option to publish the peer review history of their article (what does this mean?). If published, this will include your full peer review and any attached files.

Reviewer #1: No

---

## [Editor Report · Acceptance letter]

21 Mar 2024

PONE-D-23-14890R1 

PLOS ONE

Dear Dr. Yang, 

I'm pleased to inform you that your manuscript has been deemed suitable for publication in PLOS ONE. Congratulations! Your manuscript is now being handed over to our production team.

Kind regards, 

on behalf of

Professor Nenad Filipovic 

Academic Editor

PLOS ONE